# Monomorphic VT Non-Inducibility after Electrical Storm Ablation Reduces Mortality and Recurrences

**DOI:** 10.3390/jcm11133887

**Published:** 2022-07-04

**Authors:** Radu Vătășescu, Cosmin Cojocaru, Alexandrina Năstasă, Sorin Popescu, Corneliu Iorgulescu, Ștefan Bogdan, Viviana Gondoș, Antonio Berruezo

**Affiliations:** 1Cardiology Department, Emergency Clinical Hospital of Bucharest, 014461 Bucharest, Romania; cojocaru.r.b.cosmin@gmail.com (C.C.); iorgulescu_corneliu@yahoo.com (C.I.); stefan_n_bogdan@yahoo.com (Ș.B.); 2Faculty of Medicine, “Carol Davila” University of Medicine and Pharmacy, 050474 Bucharest, Romania; sorinpopescu63@gmail.com; 3Cardiology Department, “Elias” University Emergency Hospital, 011461 Bucharest, Romania; alexandrina.nastasa@yahoo.com; 4Department of Medical Electronics and Informatics, Polytechnic University of Bucharest, 060042 Bucharest, Romania; viviana.gondos@hellimed.ro; 5Teknon Medical Center, 08022 Barcelona, Spain; antonio.berruezo@quironsalud.es

**Keywords:** electrical storm, catheter ablation, time-to-event, survival, recurrence, ventricular tachycardia

## Abstract

**Background**: Electrical storm (ES) is defined by clustering episodes of ventricular tachycardia (VT) and is associated with severe long-term outcomes. We sought to evaluate the prognostic impact of radiofrequency catheter ablation (RFCA) in ES as assessed by aggressive programmed ventricular stimulation (PVS). **Methods**: Single-center retrospective longitudinal study with 82 consecutive ES patients referred for RFCA with a median follow-up (IQR 25–75%) of 45.43 months (15–69.86). All-cause mortality and VT recurrences were assessed in relation to RFCA outcomes defined by 4-extrastimuli PVS: Class 1—no ventricular arrhythmia; Class 2—no sustained monomorphic VTs (mVT) inducible, but non-sustained mVTs, polymorphic VTs, or VF inducible; Class 3—clinical VT non-inducible, other sustained mVTs inducible; and Class 4—clinical VT inducible. **Results**: Class 1, Class 2, Class 3, and Class 4 were achieved in 56.1%, 13.4%, 23.2%, and 7.4% of cases, respectively. The combined outcome of Class 1 + Class 2 (no sustained monomorphic VT inducible) led to improved survival (log-rank *p* < 0.001) and reduced VT recurrence (log-rank *p* < 0.001). Residual monomorphic VT inducibility (HR 6.262 (95% CI: 2.165–18.108, *p* = 0.001), NYHA IV heart failure symptoms (HR 20.519 (95% CI: 1.623–259.345), *p* = 0.02)), and age (HR 1.009 (95% CI: 1.041–1.160), *p* = 0.001)) independently predicted death during follow-up. LVEF was not predictive of death (HR 1.003 (95% CI: 0.946–1.063) or recurrences (HR 0.988 (95% CI: 0.955–1.021)). **Conclusions**: Non-inducibility for sustained mVTs after aggressive PVS post-RFCA leads to improved survival in ES, independently of LVEF.

## 1. Introduction

Electrical storm (ES) is defined by sustained episodes of ventricular tachycardia (VT) clustering over a short interval of time. Even in survivors, its occurrence is strongly associated with severe long-term clinical outcomes [1,2]. Radiofrequency catheter ablation (RFCA) may lead to abolition of all monomorphic VTs (mVTs) in ≅ 70% of cases and improved prognosis [3,4,5]. Definitions of optimal procedural results and their expected impact on clinical outcomes are still under refinement. We aimed to characterize the procedural results of ES RFCA as assessed by a more aggressive programmed ventricular stimulation (PVS) and to evaluate their impact on clinical outcomes.

## 2. Materials and Methods

### 2.1. Study Population

We retrospectively analyzed 82 consecutive patients who were underwent RFCA for ES in our center between 2014 and 2021. The study protocol complied with the Declaration of Helsinki, and it was approved by the human research committee of the Emergency Clinical Hospital of Bucharest Ethics Committee (12521—1 April 2022).

The following criteria were used for patient inclusion [3]:Structural heart disease.RFCA for ES.Clinical follow-up for VT recurrence and mortality.≥3 distinct episodes of sustained malignant ventricular tachyarrhythmia requiring adequate therapy in implantable cardioverter defibrillator (ICD) recipients over a 24-h interval refractory to medical treatment and with no reversible triggers.≥3 distinct episodes of sustained malignant ventricular tachyarrhythmia requiring electrical shock or intravenous antiarrhythmic drugs (AADs) in patients without ICDs over a 24-h interval refractory to medical treatment and with no reversible triggers.

### 2.2. Electrophysiology Study

Patients who had a history (>7 days prior to admission) of AAD therapy-suppressed ES were considered elective ES. Patients with ES during the 7 days prior to admission who were pharmacologically stabilized and had no recurring VTs 24 h before RFCA were considered acute stabilized ES. Acute ES patients were defined by recurring VTs during the 24 h prior to RFCA despite AAD treatment.

Patients underwent RFCA in a fasting state and under conscious sedation (with subcutaneous morphine for epicardial procedures and additionally Acetaminophen/Nefopam non-opioid analgetics for persistent thoracic pain) and pre-medication with low doses of midazolam. Invasive signal analysis was performed using a dedicated electrophysiology recording and analysis system (Boston Scientific Labsystem PRO EP Recording System v.2.7.0.16). High-density electroanatomical mapping (>1800 points, 70% of points emphasizing scar and its border areas) was performed in sinus rhythm (SR) with 16–500 Hz signal filtering (CARTO-3^TM^, Biosense Webster, Diamond Bar, California). A standard transvenous 6-F decapolar steerable diagnostic catheter was placed inside the distal coronary sinus. A mapping/ablation catheter was placed into the RV via the transfemoral approach or into the LV via the transseptal or retrograde aortic approach. When required, epicardial access was obtained by fluoroscopy-guided anterior percutaneous subxiphoid puncture. Remote magnetic navigation (RMN) (Niobe II, Stereotaxis Inc., St. Louis, MO, USA) was used in the majority of cases. For selected cases, mapping was performed with multielectrode catheters.

Electrogram signal analysis was performed similarly to previous communicated procedural protocols and criteria [3,6,7,8,9]. Normal myocardium was defined by endocardial bipolar signals amplitude > 1.5 mV, LV unipolar signals amplitude > 8.3 mV, RV unipolar signals amplitude > 5.5 mV, and epicardial bipolar signals amplitude > 1 mV while dense scar and borderline myocardium were defined by < 0.5 and 0.5–1.5 mV, respectively (Figure 1).

### 2.3. Radiofrequency Catheter Ablation

Ablation strategy consisted of initial scar-dechanneling by elimination of conduction channels scar entrance (CCE) [10] with RF lesions by open-irrigated ablation catheters (35–50 watts, 45 °C). CCE RFCA was performed until confirmation of scar dissociation and/or scar exit-block. Exceptions from the initial scar dechanneling strategy were hemodynamically tolerated VTs, which were repeatedly mechanically induced during mapping, for which activation and/or entrainment mapping was performed.

After confirmation of CCE RFCA by scar dissociation and/or exit-block end-procedural, the programmed ventricular stimulation (PVS) protocol was performed. The PVS protocol was routinely performed with at least 2 drive cycle lengths (CLs) and 4 extrastimuli (ESx) (3 ESx in patients with severe periprocedural heart failure (HF) symptoms or extreme frailty) (at a minimum of 200 ms or until ventricular refractoriness was reached) from 2 sites of the scar border area (usually medially and laterally to the scar) to assess for VT inducibility.

If mVTs were still inducible after initial scar-dechanneling, scar reconnection was tested and if confirmed, substrate-mapping and CCE RFCA were repeated. If scar reconnection was not present, PVS was repeated. Subsequently, activation/entrainment mapping was performed in hemodynamically tolerated induced VTs or evaluation for another substrate was performed and if demonstrated, scar-dechanneling was consequently performed. If activation-guided ablation was not possible due to VT hemodynamic instability and scar-dechanneling was not effective, the pace-mapping strategy [11] was used for identification of VT critical isthmus.

Procedural outcome was defined by one of the following 4 types of response to end-procedural PVS:Class 1 defined by absolute non-inducibility (no inducibility for either monomorphic VTs or for polymorphic VTs or ventricular fibrillation (VF)).Class 2 defined by monomorphic VT non-inducibility (no inducibility for sustained monomorphic VTs; however, residual inducibility for non-sustained mVTs, polymorphic VTs or VF inducible at 4 Esx PVS).Class 3 defined by residual monomorphic VT inducibility (no inducibility for clinical VT; however, residual inducibility for other sustained monomorphic VTs).Class 4 defined by procedural failure due to persistent inducibility of clinical VT.

If no VT EKG documentation was available (in ICD recipients), clinical VT abolition was considered if no VTs with a similar intracardiac electrogram (iEGM) or cycle length were inducible (±20 msec) at PVS.

For data analysis, in addition to the previously defined procedural outcomes, we separately analyzed the combined outcome of Class 1 and Class 2 (which represented the subgroup of patients with end-procedural non-inducibility for monomorphic VTs).

Post-procedural ICD were programed to allow identification of the slowest induced VT. In ES patients without ICDs who achieved absolute non-inducibility during end-procedural PVS and had no ICD primary prophylaxis indication, ICD was withheld. In ES patients without ICDs who did not achieve absolute non-inducibility during end-procedural PVS or had ICD primary prophylaxis indication, the implant was performed accordingly [12].

### 2.4. Follow-Up

All patients were monitored from the last performed ablation procedure. Follow-up data was obtained from medical records and routine periodical 6 month-interval post-RFCA ICD interrogation. For patients not evaluated in our center, telephone interviews were performed with referring physicians and patients and ICD interrogations were obtained from the referring physicians. ICD interrogation was performed in all patients who were alive in January 2022.

In ICD patients, recurrent VT/VF episodes were defined as any appropriate therapy including ATP. In patients without ICDs, recurrent VT episodes were defined as at least one documented sustained VT episode (symptomatic or on Holter recordings).

All-cause mortality rates were retrospectively analyzed during the post-RFCA monitoring interval, irrespective of cardiovascular and non-cardiovascular causes of death.

### 2.5. Statistical Analysis

Continuous data was expressed as mean ± standard deviation (SD) for normally distributed data and median (IQR; 25–75%) or median (minimum-maximum) for non-normally distributed data. Categorical data was expressed as count (percentage). The normality of data was evaluated by the Kolmogorov–Smirnov test. Categorical variables were compared using the Fisher’s exact test/chi-square analysis and continuous variables were compared using the Student t-test if normally distributed and non-parametric tests (Mann–Whitney U Test).

Survival curves were plotted via the Kaplan–Meier method and the statistical pairwise over strata comparison between curves was determined using the log-rank test. Univariate and multivariate Cox proportional hazards regression was performed to assess the prognostic power of clinically plausible event predictors. A 2-sided *p*-value < 0.05 was considered statistically significant. Statistical analysis was performed using SPSS version 23 (IBM Corp., Armonk, NY, USA) software and Prism 9 (GraphPad Software, LLC, San Diego, CA, USA).

## 3. Results

### 3.1. Patient Characteristics

The baseline features are summarized in Table 1. In total, 82.9% were males, with a mean age 58.75 ± 12.47 years. The median (IQR; 25–75%) time from RFCA to last follow-up was 45.43 months (15–69.86). There were 70 (85.4%) previous ICD recipients. NICM included arrhythmogenic cardiomyopathy (AC, *n* = 8, 9.8%), idiopathic dilated CMP (DCM, *n* = 8, 9.8%), post-myocarditis CMP (*n* = 6, 7.3%), valvular CMP (*n* = 4, 4.9%), and hypertrophic cardiomyopathy (HCM, *n* = 1, 1.21%). ICM patients were older (61.27 ± 10.66 years vs. 53.42 ± 14.44 years, *p* = 0.007) and more frequently dyslipidemic (*p* = 0.004) and smokers (*p* = 0.04). Most of the patients (*n* = 56, 68.29%) had a pre-RFCA LVEF < 35%.

### 3.2. General and Comparative Intra-Procedural Data and Procedural Outcomes

A total number of 107 ablation procedures (minimum 1, maximum 3) were performed in 82 patients, mostly in acute stabilized ES (*n* = 62, 75.6%). The general characteristics of clinical VT and the procedural data are summarized in Table 2. NICM had a trend of more numerous RFCA procedures (1.42 ± 0.57 vs. 1.27 ± 0.52, *p* = 0.267), longer procedures (232.29 ± 114.23 vs. 190.33 ± 69.81, *p* = 0.108), and more inducible VTs (2.38 ± 1.35 vs. 2.22 ± 2.42, *p* = 0.703). Fluoroscopy times were significantly longer in NICM procedures (22.68 ± 16.86 vs. 9.36 ± 6.59, *p* = 0.028). Epicardial ablation was significantly more frequent in NICM (48.14% of NICMs (*n* = 13) vs. 11.36% of ICMs (*n* = 5), *p* < 0.001). There were no differences regarding procedural time (210.35 ± 89.83 vs. 181.93 ± 81.83 min, *p* = 0.259) or fluoroscopy times (14.73 ± 13.51 vs. 9.72 ± 5.47 min, *p* = 0.475) between procedures with RMN and those without. ICMs had a trend of slower VTs than NICMs, without clinical significance (159.90 ± 39.56 ms vs. 149.67 ± 31.35, *p* = 0.328). There were 12 (14.63%) subjects with sustained monomorphic VT prior to PVS (incessant VT or mechanically induced). Activation mapping was performed in five of them (6.09%) who were hemodynamically stable while the other seven (8.53%) required cardioversion or overdrive pacing.

A Class 1 outcome was obtained in 46 (56.1%) patients (*n* = 37 defined by 4-Esx PVS and *n* = 9 defined by 3-Esx PVS). The combined outcome of Class 1 + Class 2 was obtained in 57 (69.5%) patients. ICM and NICM had similar rates of the Class 1 outcome (31 (56.4%) vs. 15 (55.6%), *p* = 0.565) and the combined outcome of Class 1 + Class 2 (40 (72.7%) vs. 17 (63%), *p* = 0.257). There were no significant differences in the age (57 ± 13.13 vs. 60.19 ± 11.61 years, *p* = 0.355) and LVEF (33.12 ± 12.18% vs. 30.57 ± 11.08%, *p* = 0.233) between patients that obtained a Class 1 outcome vs. those with other outcomes.

A total of 11 (13.41%) ICDs were implanted post-RFCA out of which 4 were cardiac resynchronization devices. There was one patient who had preserved LVEF and achieved a Class 1 outcome and was not implanted with an ICD. AADs post-RFCA were amiodarone + BB in 53 (64.63%), BB alone in 21 (25.60%), amiodarone alone in 3 (3.65%) patients, and 5 (6.09%) had other AADs. The type of post-RFCA treatment did not influence the rate of VT recurrence (*p* = 0.118) or death (*p* = 0.147) during follow-up. Particularly, post-RFCA amiodarone treatment versus no post-RFCA amiodarone treatment did not impact either recurrence (*p* = 0.072) or death (*p* = 0.140).

A total of eight (9.75%) procedural complications occurred, with three (3.63%) pericardial effusions (which did not require pericardiocentesis) and three (3.63%) vascular access hematomas (which did not require surgical intervention).

### 3.3. Post-RFCA Survival, VT Recurrences, and Predictors of Death and VT Recurrence

There were 23 (28%) deaths and 28 (34.14%) VT recurrences during follow-up. There were 7 deaths (30.43% of all deaths) and 17 recurrences (60.71% of all recurrences) during the first year post-RFCA. The median time-to-death was 15 months (range 0.7–64.4 months). The median time-to-recurrence was 6 months (range 0.2–51 months). There was similar all-cause mortality during the follow-up interval in ICM and NICM patients (14 (25.5%) vs. 9 (33.33%), *p* = 0.602). There was a trend of more frequent recurrences in NICM compared to ICM (13 (48.14) vs. 15 (27.27%), *p* = 0.081).

#### 3.3.1. Mortality Rates by RFCA Timing, RFCA Outcome, and VT Recurrence

There was no significant mortality difference in relation to RFCA timing (i.e., elective ES vs. acute stabilized ES vs. acute ES, *p* = 0.124).

Figure 2 summarizes the Kaplan–Meier survival mortality curves stratified by outcomes. Class 1 patients had better survival compared to Class 3 (log rank *p* = 0.001) and Class 4 (log rank *p* < 0.001) patients and there was similar survival compared to Class 2 patients (log rank *p* = 0.833). The combined outcome of Class 1 + Class 2 had improved survival compared to those without the combined outcome (log rank *p* < 0.001) (Figure 3).

Patients without VT recurrence had improved survival compared to those with VT recurrence (log rank *p* < 0.001) (Figure 4).

Patients with residual mVT inducibility (combined outcome Class 3 + Class 4) had similar mortality (log rank *p* = 0.107) or recurrences (log rank *p* = 0.085) irrespective of PVS protocol (4-ESx or 3-ESx).

#### 3.3.2. Recurrence Rates by Outcomes

Figure 5 summarizes the Kaplan–Meier recurrence curves stratified by outcomes. During follow-up, VT recurrence in Class 1 was similar to Class 2 (log rank *p* = 0.921) and significantly lower than in Class 3 (log rank *p* < 0.001) and Class 4 patients (log rank *p* < 0.001).

Recurrences in the Class 1 + Class 2 subgroup (monomorphic VT non-inducible) were significantly lower than the Class 3 + Class 4 subgroup (monomorphic VT inducible) (*n* = 11 (19.3%) vs. *n* = 17 (68%), *p* < 0.001).

There were two cases (7.14%) of VT recurrences by ES recurrence during the first month post-RFCA. Both patients had Class 4 outcome post-RFCA and died during hospitalization. The other 26 (92.85%) recurrence patients developed isolated VT episodes. Three of these (11.53%) did not recur after oral amiodarone initiation and two (7.69%) were associated with HF episodes and did not recur after acute HF decompensation control. The rest of the recurrence patients (*n* = 21 (75%)) manifested isolated arrhythmic episodes, which were treated conservatively.

#### 3.3.3. Predictors of Post-RFCA Death

Multivariate Cox regression for post-RFCA mortality (Table 3) showed that residual inducibility of sustained mVTs (HR 6.262, 95% CI 2.165–18.108), NYHA IV class upon admission (HR 20.519, 95% CI 1.623–259.345), and age (HR 1.099, 95% CI 1.041–1.160) were independent predictors of death. LVEF was not predictive of death (HR 1.003. 95% CI: 0.946–1.063) or recurrences (HR 0.988 (95% CI: 0.955–1.021). Additionally, there were no significant LVEF differences in survivors vs. non-survivors (33.57 ± 12.57 vs. 29.43 ± 9.09, *p* = 0.160) or recurrence vs. non-recurrence subgroups (31.27 ± 10.18 vs. 32.46 ± 12.34, *p* = 0.676).

## 4. Discussion

### 4.1. Even with a More Aggressive PVS Protocol Non-Inducibility of Residual Sustained mVTs Is Achievable by RFCA in Almost 70% of Cases

This observation is consistent with previous studies [3,4,5]. Similar rates of Class 1 and the combined outcome Class 1 + Class 2 were achieved in ICM and NICM, which is consistent with existing data [4,13]. However, procedural difficulty was increased in NICM, requiring more frequent epicardial RFCA due to the greater subepicardial distribution of the relevant substrate [4,13,14] and longer fluoroscopy times.

Our analysis did not reveal differences between the RFCA timing scenarios, most likely caused by the limited number of cases and imbalance between groups (75.6% being acute stabilized ES), although large-scale data shows that emergency RFCA without prior HF and electrical stabilization leads to worse outcomes [15].

It is commonly known that ES presentation is dominated by monomorphic VTs. Multiple periprocedural induction, especially of faster VTs, may compromise RFCA safety by mandating repeated electrical cardioversions/defibrillations and progressive hemodynamic deterioration. The observed mean mVT rate of 156.06 ± 36.95 ms is within the range of previously reported data with a trend of faster VTs in NICM patients [4,16].

In this sense, it is to be emphasized that employing a baseline strategy of scar-dechanneling in SR has substantially improved safety in larger datasets precisely by limiting the induction of fast VTs (less than 1% hemodynamic deterioration vs. previously reported 11% rates in activation mapping-based ablation) [16,17]. As a possible consequence of this, Muser et al. [4] showed that VT CL did not influence either recurrence or death/transplantation during follow-up (precisely by avoiding the deleterious effect of fast VTs on procedural outcomes). Secondly, this may also explain why none of our subjects required mechanical support.

A more aggressive end-procedural PVS with 4 ESx was utilized in our study. The most commonly used PVS protocol for end-procedural VT inducibility testing in previous large-scale publications involves up to 3 ESx from two distinct anatomical sites [3,4].

Limited data is currently available concerning the impact of exclusively 4 ESx-induced ventricular arrhythmias, whether it is a non-specific response to aggressive stimulation (17.45% (*n* = 10) out of 57 subjects tested with 4-ESx were assigned to Class 2 versus 4% (1 out of 25) of those tested by 3-ESx PVS) or it carries prognostic information. Recent data showed that although residual inducible non-sustained VTs (NSVTs) (both pVTs and mVTs) were independently associated with a higher 3-fold risk of VT recurrence, patients with NSVTs induced by 4 ESx did not manifest VT recurrences [18]. While emphasizing the reduced number of patients, we observed 4 ESx-inducible NSVTs (either pVT or non-sustained mVTs) or VF (i.e., Class 2 outcome) did not lead to either excess recurrences or mortality when compared to Class 1. Consequently, we performed a separate analysis of the combined outcome of Class 1 + Class 2 (no sustained monomorphic inducible VT), which clearly improved prognosis compared to the combined outcome of Class 3 + Class 4 (residual sustained monomorphic VTs).

Furthermore, the majority of patients with sustained VTs induced by 4 ESx had subsequent recurrences [18]. Similarly ischemic patients with inducible VTs by 4 ESx have been shown to have equivalent arrhythmic risk as those with VTs induced by 3 ESx [19]. Recently, it has been shown that up to 17% of inducible VTs may be missed if 4 ESx PVS is not utilized and, moreover, patients exhibiting Class 1 after 4 ESx PVS may have better clinical outcomes compared to those achieving Class 1 by 3 ESx PVS protocols post-RFCA [20]. Similarly, in our study, there were eight Class 3 patients and two Class 4 patients (i.e., *n* = 10 patients with residual mVT inducibility, 12.2%) who only became apparent during 4-ESx testing, and it is not entirely surprising that these patients seem to have a similar prognosis as those inducible by 3-ESx, which reinforces the potential benefit of more aggressive testing.

Absolute non-inducibility (i.e., Class 1 outcome) is associated with improved clinical outcomes in ES patients undergoing RFCA. The benefit of absolute non-inducibility achieved by RFCA over long-term survival was first documented in meta-analyses and a large-scale cohort study of ischemic VT patients [21,22]. This was subsequently validated by the results of Vergara et al., which demonstrated the survival and VT recurrence benefit were lower in Class 1 patients undergoing RFCA for ES in comparison to those with residual inducible clinical or non-clinical VTs [3]. There are notable differences between studies addressing the incremental survival benefit of exclusive clinical VT elimination (and not other inducible VTs) while the latter (in ES context) stated better survival by Class 1 in comparison to solely clinical VT abolition [3,21], in accordance with our data. Furthermore, VT recurrence after RFCA led to lower survival of ES patients, which is consistent other studies [3,23]. Provided that limited data was available concerning the specific modality of recurrence due to the retrospective nature of this study, in a subset of 24 “Class 1” patients (52.17%), there was no recurrence of clinical VT or ES but by non-clinical VTs (3 cases). Even if absolute non-inducibility is achieved by RFCA, subsequent VT recurrences may also be possible due to the dynamic nature of time-dependent scar reverse-remodeling, periprocedural edema, antiarrhythmic treatment discontinuation, periprocedural anesthesia effects, or autonomic disturbances or novel structural disease [24,25].

There were no differences in post-RFCA survival or VT recurrence in ischemic versus non-ischemic patients. This is a similar conclusion to that of the larger-scale observational study of Muser et al. [4] and is contrary to initial beliefs [23,26] that indicated a worse long-term prognosis (all-cause mortality and recurrent arrhythmias) in non-ischemic vs. ischemic ES patients. Considering that the former are younger and have less comorbidities, we hypothesize that if a Class 1 outcome is achieved, future survival and arrhythmic prognosis is highly dependent on the specific rate of disease progression.

Interestingly, the “no-PVS” subgroup may have a similar rate of mortality as patients with residual monomorphic VT inducibility [3]. Even so, we hypothesize these patients may not be phenotipically identical to “Class 3” subjects (who may exhibit more complex substrate rather than severe comorbidities and advanced HF). In these frail patients, delayed non-invasive PVS may be of interest, with several studies showing improvements in post-procedural assessment of prognosis and the need for re-ablation, while avoiding the risk of periprocedural decompensation, confounding effects of anesthesia, post-PVS regressing ablation lesions, and acute autonomic disturbances [27,28].

### 4.2. Influence on NYHA Class and Age vs. Lack of Influence of LVEF on Death and Recurrences

In addition to residual sustained mVT inducibility, NYHA class IV symptoms at admission and older age independently predicted death during follow-up. PAAINESD [29] and I-VT [30] scores have previously demonstrated incremental mortality risk of acute HF and of advanced age. However, we believe it is critical to distinguish between two HF clinical phenotypes, which may further aid risk stratification: ES-induced HF (in a previously stable patient with better prognosis if ES abolition is obtained) and HF-induced ES (which is associated with highest mortality).

Different from other studies, recurrence and survival were not associated with better LVEF. Although this might be explained by the limited number of patients analyzed retrospectively and to underestimation of LV contractile function due to stunned but viable myocardium existing in some patients, one cannot exclude that a more aggressive PVS protocol might have exposed residual arrhythmogenic substrate that was eliminated by ablation even in patients with severely depressed LVEF, with further reduction in recurrences and mortality. Further support for this idea stems from recent studies showing that LVEF did not maintain its predictive power when substrate arrhythmogenicity was assessed by cardiac MRI [31].

### 4.3. Limitations

There are specific limitations of this study. Firstly, despite similar evaluation protocol, this was a limited study population with retrospective data acquisition. Correspondingly, the impacts of patient characteristics and procedural outcomes were only analyzed in relation to all-cause mortality, irrespective of specific cardiovascular and non-cardiovascular causes of death. Furthermore, there is inter-individual variability regarding previous and post-AAD treatment, which may impact long-term outcomes. The majority of patients received post-RFCA treatment with amiodarone (67.1%), which may impact comparative data analysis with other treatment subgroups, especially in a limited study population. However, recent large-scale observational studies have shown a debatable effect of AADs on VT recurrence after ES RFCA [4]. There was a predominance of RMN-guided procedures, which may impact comparison to those without.

Secondly, 20.73% (*n* = 17) of patients had no VT EKG documentation prior to RFCA, for whom outcome class definition was strictly based on ICD-derived interpretation, which may erroneously assign cases between Class 3 and Class 4 outcomes. Consequently, we performed a combined outcome analysis (Class 1 + Class 2 vs. Class 3 + Class 4), and we exclusively included the combined outcome dichotomization into the regression analysis.

Although a modified strategy of PVS-site selection inside the scar border-zone was used (which may limit comparability to standard RV-pacing-based protocols), we believe that this personalized PVS may unveil residual arrhythmogenic substrate at least as well as standard stimulation sites.

We acknowledge that 4 ESx PVS was performed in only 69.51% (*n* = 57) of subjects, which can impact the clinical significance of this protocol. However, most of the patients not tested with 4-ESx had mVT inducible with 3-ESx (*n* = 15, 18.29%), thus making 4-ESx PVS obsolete. Additionally, 10 patients (12.20%) were tested only with 3-ESx PVC due to their frailty, which is in range with other larger-scale “3 ESx” studies that have reported refraining completely from end-procedural PVS in 5–10% of patients [3,4].

## 5. Conclusions

The abolition of all inducible sustained mVTs by catheter ablation as assessed by a more aggressive PVS protocol led to more favorable clinical outcomes in ES patients independently of LVEF.

## Figures and Tables

**Figure 1 jcm-11-03887-f001:**
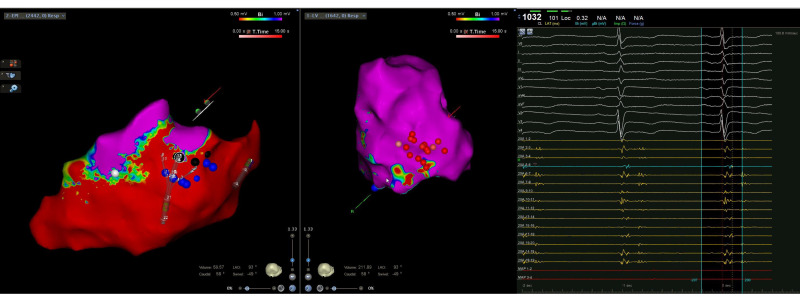
Intraprocedural CARTO three-dimensional left ventricular epicardial (**left**) and endocardial (**right**) electroanatomical multielectrode mapping displaying a predominantly epicardial inferior and inferolateral wall scar with limited endocardial involvement. Initial endocardial radiofrequency applications (red dots) were not effective in eliminating all the conduction channel entries (blue dots) observed during mapping.

**Figure 2 jcm-11-03887-f002:**
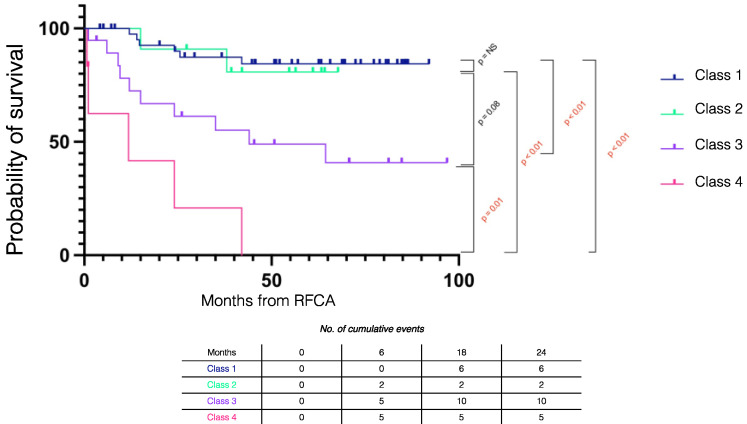
Kaplan–Meier survival curves stratified by procedural outcome in electrical storm patients. RFCA = radiofrequency catheter ablation.

**Figure 3 jcm-11-03887-f003:**
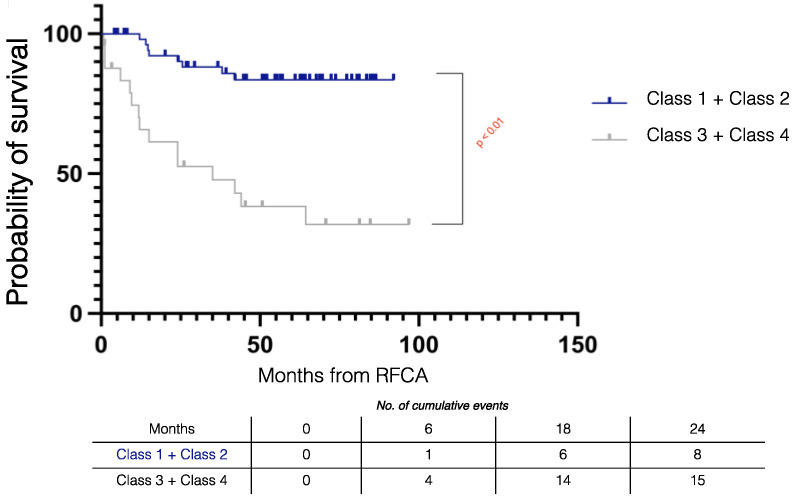
Kaplan–Meier survival curves stratified by combined procedural outcomes in electrical storm patients. RFCA = radiofrequency catheter ablation.

**Figure 4 jcm-11-03887-f004:**
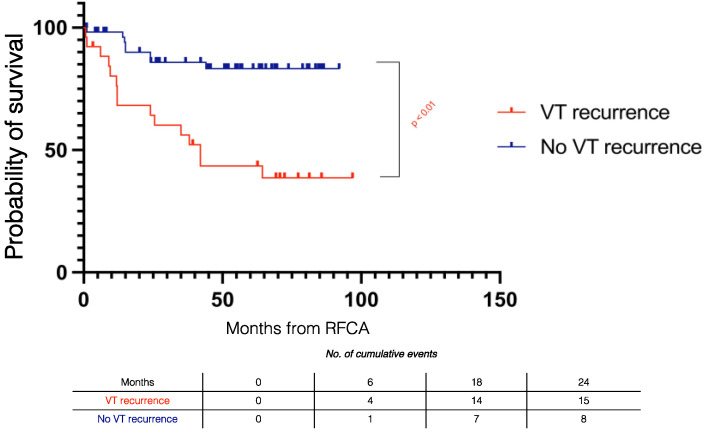
Kaplan–Meier survival curves stratified by VT recurrence during follow-up in electrical storm patients. RFCA = radiofrequency catheter ablation.

**Figure 5 jcm-11-03887-f005:**
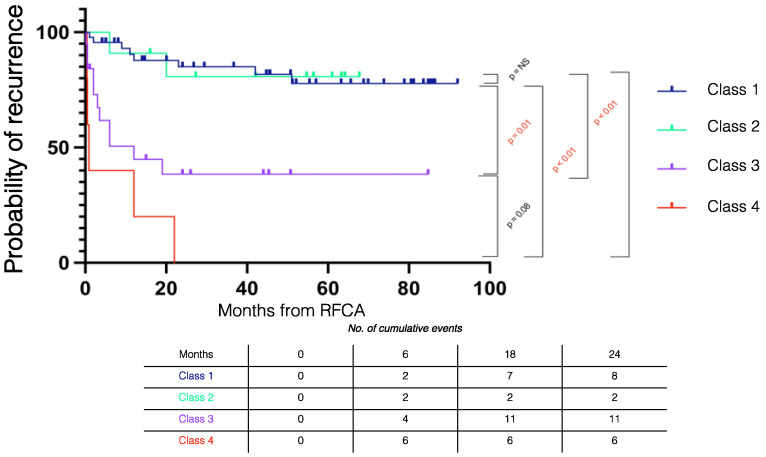
Kaplan–Meier recurrence curves stratified by procedural outcome. RFCA = radiofrequency catheter ablation.

**Table 1 jcm-11-03887-t001:** Characteristics summary of ES patients referred for RFCA. PCI = percutaneous coronary intervention, CABG = coronary artery bypass graft surgery, ES = electrical storm, ICD = internal cardioverter defibrillator, LVEF = left ventricular ejection fraction, RFCA = radiofrequency catheter ablation.

* **Demographic and Clinical Data** *	
Age (years)	58.75 ± 12.47
Male gender	68 (82.9%)
Hypertension	45 (54.87%)
Diabetes mellitus	18 (21.95%)
Dyslipidemia	46 (56.09%)
Chronic kidney disease	21 (25.60%)
Obesity	22 (26.8%)
Smoker	18 (21.95%)
Ischemic cardiomyopathy	55 (67.1%)
History of PCI	28 (34.1%)
History of CABG	7 (8.5%)
NYHA I	11 (13.4%)
NYHA II	44 (53.7%)
NYHA III	22 (26.8%)
NYHA IV	4 (6.1%)
ICD recipient prior to ES episode	70 (85.36%)
*ES episode characteristics*	
Acute ES	5 (6.1%)
Acute stabilized ES	62 (75.6%)
Elective ES	15 (18.3%)
Time from ES to RFCA (weeks)	1.66 ± 3.18
ICD therapies (ICD recipients) (minimum-maximum)	5 (3–103)
ICD shocks (ICD recipients (minimum-maximum)	4 (2–86)
External shocks (non-ICD recipients) (minimum-maximum)	4 (3–9)
*Pre-ES treatment*	
Amiodarone + beta-blocker	41 (50%)
Beta-blocker alone	31 (37.8%)
Amiodarone alone	6 (7.31%)
Sotalol	4 (4.87%)
Amiodarone i.v. on admission	52 (63.4%)
*Two-dimensional echocardiography data*	
LVEF (%)	32.32 ± 11.72
End-diastolic LV diameter (IQR; 25–75%) (mm)	58 (51–65)
RV diameter (IQR; 25–75%) (mm)	37 (32-43)
TAPSE (IQR; 25–75%) (mm)	20 (17.25–21.75)

**Table 2 jcm-11-03887-t002:** Clinical VT characteristics, procedural characteristics, outcomes, and complications. VT = ventricular tachycardia, RBBB = right bundle branch block, ES = electrical storm, no. = number, IQR = interquartile range (25–75), FAM = fast anatomical mapping, LV = left ventricle. * Out of the 25 subjects with no PVS-4 Esx testing, 4 subjects had a Class 4 outcome and 11 subjects had a Class 3 outcome during initial 3-ESx PVS; thus, 4-ESx PVS was not subsequently performed; the remaining 10 subjects with no 4-ESx PVS (nine subjects with Class 1 outcome and one with Class 2 outcome) were assigned to procedural outcomes only by 3-ESx due to frailty; 37 patients were assigned to Class 1 and 10 subjects to Class 2 by 4 ESx PVS; moreover, there were 8 Class 3 patients and 2 Class 4 patients who were discovered at 4-ESx PVS.

*Clinical VT Characteristics*	
mVT rate (bpm)	156.06 ± 36.95
QRS duration DII (ms)	175.16 ± 35.74
Shortest RS interval (ms)	132.67 ± 44.77
RBBB-like morphology	49 (59.8%)
ES without ECG documentation of VT	17 (20.7%)
* Procedural characteristics *	
Mean no. of procedures	1.32 ± 0.54
Mean days of hospitalization	10.51 ± 10.74
4 ESx PVS *	57 (69.51%)
Remote magnetic navigation	63 (76.80%)
Transseptal approach	67 (81.70%)
Epicardial approach	18 (22%)
Substrate ablation	75 (91.46%)
Activation mapping	5 (6.09%)
Mean no. of VTs induced	2.28 ± 2.12
Median no. of ablation points (IQR; 25–75%)	34.5 (21–55.25)
FAM endocardial mapping points (IQR; 25–75%)	1756 (1081.25–2211.75)
Multielectrode catheter mapping	8 (9.8%)
Median procedural time (IQR; 25–75%) (mins)	176 (145–246)
Median fluoroscopy time (IQR; 25–75%) (mins)	10.2 (5.85–21.55)
Median LV endocardial volume (IQR; 25–75%) (mL)	222.80 (201.45–294.52)
* Procedural outcomes *	
Class 1	46 (56.1%)
Class 2	11 (13.4%)
Class 3	19 (23.2%)
Class 4	6 (7.4%)
* Procedural complications *	
Stroke	1 (1.21%)
Pericardial effusion	3 (3.63%)
Arterial embolism	1 (1.21%)
Vascular access hematoma	3 (3.63%)

**Table 3 jcm-11-03887-t003:** Univariate and multivariate Cox regression analysis for prediction of survival without death in post-RFCA ES patients. ES = electrical storm, RFCA = radiofrequency catheter ablation, mVT = monomorphic ventricular tachycardia, LVEF = left ventricular ejection fraction.

Variable	Univariate Analysis		Multivariate Analysis	
	HR (95% CI)	* p * -Value	HR (95% CI)	* p * Value
MVTI or FAIL	5.947 (2.516–14.055)	<0.001	6.262 (2.165–18.108)	0.001
NYHA class I	Reference		Reference	
NYHA class II	1.249 (0.270–5.783)	0.776	1.288 (0.213–7.785)	0.783
NYHA class III	3.577 (0.767–16.678)	0.105	4.769 (0.714–31.851)	0.107
NYHA class IV	7.389 (1.218–44.821)	0.030	20.519 (1.623–259.345)	0.02
Age	1.063 (1.018–1.110)	0.006	1.009 (1.041–1.160)	0.001
mVT recurrence	4.611 (1.952–10.893)	<0.001	1.803 (0.670–4.853)	0.243
LVEF	0.968 (0.933–1.005)	0.087	1.003 (0.946–1.063)	0.924

## Data Availability

Study data is available upon request by any third parties.

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
