# Peer review of "Monomorphic VT Non-Inducibility after Electrical Storm Ablation Reduces Mortality and Recurrences"

_jcm, 2022, doi:10.3390/jcm11133887_

Round 1

Reviewer 1 Report

Vătășescu et al performed an analysis about the role of an aggressive programmed ventricular stimulation after RF catheter ablation in patients with electrical storm. They stated that the abolition of all inducible sustained VTs led to more favourable clinical outcomes. The paper is well written and clear and describe the role of ventricular stimulation in these patients. The authors should be congratulated for this report. I have just some comments:

a) The authors should be clarified the impact of AAD therapy in the long term follow-up.

b) A great number of patients had ICD but the authors didn't report any information about ICD therapy.

c)   Paragraph 3.3 should be shortened and data presented more clearly.

d) In order to improve graphical presentation, the authors may added some CARTO map (unipolar and/or bipolar voltage map, activation map, the sites of RF delivery and more other).

e) During follow-up there were 17 recurrences; please describe the management of these patients. 

Author Response

Dear Sir/Madam,

We have uploaded the response as a Word file.

We are thankful for your proposed revisions and we hope we have improved the manuscript in accordance to your recommendations.

With respect,

The Authors

Reviewer 2 Report

See uploaded file

Author Response

(The authors gave the same response as above.)

Round 2

Reviewer 1 Report

I thank the authors for the valuable responses.

Author Response

Dear Reviewer 1,

We respectfully thank you for your contribution to this manuscript's improvement.

We have attached the response to the Academic Editor's further remarks.

Best regards,

The Authors
